# Gender Difference in Type 1 Diabetes: An Underevaluated Dimension of the Disease †

**Patrizio Tatti * and Singh Pavandeep** 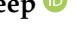

Istituto I.N.I.–Via S. Anna SNC, Grottaferrata, 00046 Roma, Italy; pavanmaan97@gmail.com
* Correspondence: info@patriziotatti.it
† Proceedings from "Gender differences in diabetes" held in Olbia, 4–5 December 2020.

**Abstract:** Gender difference in all fields of medicine and biology has recently become a topic of great interest. At present, most studies report gender differences in their secondary analysis; however, this information receives scant attention from clinicians, and is often overwhelmed by press trumpeting the overall main positive results. Furthermore, and more importantly, any statistical evaluation of results obtained without specific and careful planning in the study for the topic of research is probably worthless. There are few studies in animals, but these are not typically useful because of the different biology, pharmacodynamics and pharmacokinetics compared to humans. Type 1 diabetes is a disease where gender difference can be easily evaluated. Irrespective of the cause of the loss of pancreatic beta-cell function, the common denominators of all forms of type 1 diabetes are the absence of circulating insulin and a reduction in peripheral insulin sensitivity leading to exogenous injections being required. Consequently, exogenous insulin infusion, with any of the widely used research tools, such as the insulin–glucose clamp, can be easily used to evaluate gender difference. Female patients with type 1 diabetes have many factors that impact glucose level. For example, the hormones that drive the ovulatory/menstrual cycle and the connected change at the time of the menopause have a role on insulin action; thus, one should expect great research emphasis on this. On the contrary, there is a dearth of data available on this topic, and no pump producer has created a gender-specific insulin infusion profile. Patients are usually approached on the basis of their diagnosis. This review is intended to focus on personalized treatment, more specifically on gender, according to the modern way of thinking.

**Keywords:** gender; diabetes; type 1

## 1. Introduction

Type 1 diabetes has many different genetic and environmental causes, and a common pathophysiological ground represented by the almost total or total absence of insulin. Hyperglycemia is the cause of complications in almost all the organs of the body. The appearance, degree, and location of these complications may depend on a number of conditions. Among them are the degree of blood sugar control, the duration of the disease, the age of the patient, the type of insulin used for replacement, the number of injections, the spectrum of residual endogenous insulin secretion, the sensitivity of the peripheral tissue to the insult, the blood glucose swings (glucose variability), and not least the sexual and genetic environment and hormonal patterns. Among these conditions, sex is the simplest to control, although there are not many studies that account for this variable.

Males and females were not created equal. Gender differences are much deeper than the reproductive/sexual dimorphism, and span most of the organs.

Unfortunately, for whatever reason, these differences have been too easily dismissed in medicine and in the socio/economic environment.

We know that the differences involve physiology and also pharmacokinetics and pharmacodynamics. This mini review deals with the gender differences in type I diabetes. It is clear that biological differences play a role, but where, when and how is not clear.

## 2. Incidence

The incidence of type 1 diabetes varies across latitude, race, and economic conditions. A study in Sweden spanning from 1983 to 2002 found an annual incidence rate of 16.4/100,000 for males and 8.9/100,000 for females [1]. The differences decreased slowly with age, but the prevalence remained higher in men with an overall male/female ratio of 1.8. Notably, temporal trends and seasonal patterns (higher in January–March and lower in May–July) were the same in both sexes, thus pointing the focus to gender rather than exposure to environmental factors.

Another study [2] found an increased risk of T1DM (type 1 diabetes mellitus) in boys with neonatal infections (OR = 2.42, (CI 1.14–5.15) $p$ = 0.02) and a decreased risk in girls with neonatal infections (OR = 0.4 (CIU 0.12–1.38) $p$ = 0.15). This difference remained unaltered after removing the confounders. Diabetes is a T-cell-mediated autoimmune disease, so the difference may reside in gender-linked susceptibility and response to infections. Many papers have demonstrated sexual dimorphism in the immune cell count and activity between males and females, likely dependent on the sex hormones.

Although the point has been dissected according to age and other characteristics, at present there is no unifying view of the gender gap in type 1 diabetes. A study in Germany dealing with self-assessed adherence to prescriptions reported that poor glycemic control was found in 19% of men and in 18% of women [3].

Contrary to other organ-specific autoimmune disorders, T1DM does not show a female prevalence, with a limited roughly 45% figure [4]. However, the matter is more complicated than this. The populations with the highest prevalence of T1DM show male excess; the reverse is true for populations with low prevalence of the disease. Unfortunately the diagnosis of type 1 diabetes is not always so lear-cut. Type 1 diabetes is a distinctive disease only in children, where it can be diagnosed with utmost confidence. The appearance in these subjects is abrupt and noteworthy. Type 1 DM appears also in advanced age, over 40 years, but the disease often goes undiagnosed and confused with insulin-treated type 2 DM (type 2 diabetes mellitus). In these subjects, the HLA (human leukocyte antigen) type, insulin deficiency and insulin autoantibody changes are less evident than in children [5], or may be absent [6]. Accordingly, it is difficult to draw firm conclusions from studies in this population.

Accordingly, a careful review of prevalence identified a slight excess of males in European countries and a female prevalence in populations of Asian and African origin [7].

When we restrict the observations to those under 15 years of age, we find that in populations with an incidence rate of T1DM over 23/100,000 there is male preponderance, while the reverse is true for those populations with an incidence rate under 4.5/100,000. Another unexpected finding is that in those patients with no islet antibodies there is a strict male predominance [8]. It is difficult to draw firm conclusions from these partially conflicting data.

## 3. Clinical Aspects

Another study from Edinburgh with no clear-cut age limit reported that women were more likely to seek medical help and feel symptoms. This review indicates that male diabetics are observed to be living more effectively with diabetes, with lesser depression and anxiety and more energy and better wellbeing [9], although diabetic men with advanced disease may experience the phenomenon of erectile dysfunction, which causes serious depression and the tendency to abandon the cure. A similar sex dysfunction with vaginal dryness probably occurs in women, but this has not been investigated in depth. Furthermore, the more sophisticated sex–arousal mechanism of the female body, with relevant psychic influences, makes the topic more complicated.

Another study found that girls may be more likely to hide non-adherence to the prescription of insulin, and consequently may have feelings of guilt and self-blame [10]. This may be in part to do with the phenomenon known as "cheating", which means that girls and young women know that hyperglycemia in diabetes induces the loss of weight,

so they voluntarily reduce the insulin dose, even more so in these times of social media exposure where models of fitness and beauty are inspiring the youth.

There are very few data available on insulin action on each sex in type 1 diabetes. We can deduce some useful information to shed light on Type 1 diabetes from the data of men and women patients with type 2 diabetes. We think that some information obtained from a large number of men and women with type 2 diabetes, which is a much more metabolically complex disease, can partially be used to understand the gender differences in Type 1: (i) women appeared to be more sensitive to insulin in regard to glucose metabolism (both in the liver and in muscle); (ii) there were no differences in insulin action on lipolysis in men and women; (iii) the data available on the regulation of triglyceride and protein metabolism by insulin in men and women were too scarce to draw firm conclusions; (iv) there might be differences in the insulin-sensitizing effects of exercise and weight loss in men and women [11].

Furthermore the presence of type 1 diabetes is no guarantee against the appearance of insulin resistance in those affected. Insulin resistance in type 1 can be induced by an increased fat tissue mass due to over-treatment with insulin. Thus, some of the metabolic defects in insulin-resistant type 2 diabetics may also be present in type 1. The role of insulin has been traditionally studied in the context of glucose metabolism. However, insulin also plays a role in the inhibition of adipose tissue lipolysis and fatty acid release into the blood stream [12], regulates hepatic lipid metabolism [13], and inhibits protein breakdown [14]. In addition, insulin action and insulin resistance are not the same in any organ of the body, and we only know a little part of this [15]. It is evident that studying these differences in the intact human body is quite difficult at present, and more so in men and women separately. We know that fasting plasma glucose and glucose production and disposal are apparently not different between sexes [16], although in some studies women exhibited a greater and more prolonged suppression of endogenous glucose production rates than men [17]. This observation, if confirmed, may have great relevance in the management of insulin-induced hypoglycemia. Estrogens that are prevalent in the first part of the menstrual cycle have divergent effects on insulin action, that overall seem to be positive by increasing insulin action on peripheral tissue. Greater rates of glucose disposal during the follicular compared to the luteal phase of the menstrual cycle have been observed during a hyperglycemic (blood glucose > 200 mg/dL) hyperinsulinemic clamp [18]. Progesterone, on the other hand, seems to interfere negatively with insulin action, thus apparently increasing the need for injected insulin in type 1 DM [19].

Other hormones also play a role in this gender difference. Treatment with testosterone reduces insulin-mediated glucose disposal in women [20] and hyperandrogenemia might be the major culprit for insulin resistance in women with polycystic ovary syndrome.

It may seem absurd, but, to our knowledge, no infusion profile that accounts for these subtle cyclic changes in women on CSII (continuous subcutaneous insulin infusion) has been put forward.

There are no clear differences between men and women in terms of insulin action on lipolysis [21], although this should have been viewed in the context of different muscle mass. Finally, overall available data confirm that the response of adipose tissue lipolysis to insulin appears to be the same in the two sexes.

There may be differences in the regulation of plasma triglyceride concentration and protein metabolism by insulin and in changes in insulin action in response to hormonal stimuli. Women appear to be more sensitive to the suppression of VLDL (very-low-density lipoproteins) by insulin [22].

The available results about muscle protein synthesis raise perplexity because, ironically, the papers that do find differences report a greater rate of muscle protein synthesis in women than in men [23]. This was not anticipated because of the well-established anabolic effects of testosterone. Clearly a lot more work must be completed in this field. Some recent data show that the interest to close this gender gap is increasing. This is absolutely

necessary because the menstrual cycle with its hormone-driven swings imposes a serious challenge on women, who also have to struggle with subtle disturbing symptoms.

A study at the Mount Sinai Hospital on 16 menstruating women on an insulin-sensor-controlled automated pump, showed that the TIR (time in range) could be improved to 69% in the menstrual phase, 67% during the luteal phase, and 69% during the rest of the cycle. This may affect the lives of women at the fertile age that usually strive to cope with a high blood sugar level during their menstrual cycle. This is just a starting point, but with improved attention to the phenomenon, it is likely that the companies producing these pumps will come out with an algorithm able to create a personalized insulin delivery profile.

## 4. Conclusions

The gender gap in type 1 diabetes is far from closed. At present, we are still without a clear picture of the biology of blood glucose control in males versus females. In the case of females, we do not know enough about what happens at different ages—prepubertal, post-pubertal reproductive, and after the menopause—and what drives any change. This information is probably relevant to achieve an effective cure for patients with diabetes mellitus. We have known for a long time that the net effect of insulin on the body is the result of pharmacodynamics (effect of insulin on the body) and pharmacokinetics (effect of the body on insulin). So far, we know more on pharmacodynamics, but very little on pharmacokinetics, especially in women. We need to shed light on this to achieve any effective cure.

In conclusion, we have no firm data on gender difference in type 1 diabetes mellitus. Males appear to be more frequently affected by the disease, although this may vary in different populations, and females appear to show higher sensitivity to insulin. This aspect may be of relevance in insulin treatment and in cases of emergency treatment for hypoglycemia. Accordingly, treatment with male hormones appears to increase insulin resistance in women, and this hormonal effect may explain the lower insulin sensitivity in men. Hopefully, future studies will help to tailor a better insulin treatment in both sexes.

## 5. Patients

This section is not mandatory but may be added if there are patients resulting from the work reported in this manuscript.

**Funding:** This research received no external funding.

**Conflicts of Interest:** The author declares no conflict of interest.

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
