# Peer review of "Gender Difference in Type 1 Diabetes: An Underevaluated Dimension of the Disease"

_diabetology, doi:10.3390/diabetology3020027_

Round 1

Reviewer 1 Report

Type of manuscript: Mini Review

Title: Gender Difference: A Forgotten Dimension Of Medicine and Biology

Journal: Diabetology

Manuscript ID: diabetology-1701422

In the current review article, the authors try to give insight into gender differences in relevance to Diabetes mellitus (DM) disease in humans. In the review author describe gender difference as a crucial factor for analyzing differences and patterns in diabetes mellitus, especially Type 1 diabetes, and to describe this variation examples from type 2 diabetes cases were mentioned too in the review. Diabetes is a group of diseases and several people are affected by this disease. The author is more concerned that ignoring these differences could be disadvantageous for pharmaceutical companies. The author suggests that this gender-based study might help design proper infusion pumps by pharmaceutical companies and ultimately this will lead to great treatment therapy for diabetic patients in order to control their sugar levels and keep a healthy lifestyle.

This kind of analysis will be really helpful in pointing out key differences between both genders and ultimately provide more information and noble findings in male and female diabetic patients. However, this review needs proper structure in providing information, and it's very confusing which key points are crucial as per the authors and which are just observations.

Major comments:

Right from the title itself which says ‘’Gender Difference: A forgotten Dimension Of Medicine and biology’ gives no information stating the review is about Diabetes disease. The title itself should be strong enough to give an overview of the research work. However, the author has mentioned this only in the abstract. Another thing in the abstract is surprising that nowhere mentioned any other diabetes types except Type 1 diabetes. Nevertheless, type 2 diabetes cases were mentioned in the review to compare with type 1 cases. It is more confusing, as, in the beginning, it looked like the review is about diabetes disease. The author should properly clarify this statement in order to understand whether he is giving this statement about all the types of diabetes or specifically Type 1 diabetes.

The author should have made different chapters instead of mixing all the information from the original articles. In my view, this review lacks a lot of information and also a structure presenting data in order to understand what are the outcomes and what are the key points and what should be taken into consideration. The major thing is there is no background given for disease, only incidence, and some cases. It would have been great if the author would have explained what others have done and what was the results and what is their interpretation of this work, instead of putting all the statements together. 

Another thing sometimes author says that type1 and type2 are different and data cannot be translated in order to understand gender differences in type1, on the contrary in some paragraphs author mentioned that most of the data analyzed from type2 cases cannot be transposed to type 1DM, which is contradictory. Last but not least no continuity in statements, there are some statements randomly placed with no heading or subheading is given. I appreciate their efforts putting all the inputs but it needs a lot of work to bring in a presentable way.

Minor comments: This review needs to check by a native English speaker for grammatical mistakes, typos, writing errors, also for the proper construction of words.

Mistakes by Line numbers

Line 1 : Titel: need mention of disease

Line 4: It's not gender differensism there is no word in the dictionary please change gender differences

Line 11: Typo error ‘ not recenrtl but recently

Line 12: not difference bit differences and in last sentence however,

Line 13: typo  i is missing in information

Line 14: Not this is in part but This is the part pushed by the drug companies

Line 15: in an in-depth analysis please correct like this, Furthermore typos and space between words

Line 16: instead important importantly right word

Line 17: sentence not constructed grammatically, spelling error pharnacodynamic

Line 19: remove the

Line 20: beta-cell

Line25: correction connected

Line26: starting sentence is incomplete, in the end, full stop and comma

Line27: on the contrary, the comma after contrary, reduce space 

Line 28: there is dearth of data available missing, specific spelling correction

Line 33-34: reduce the gap between

Line35: full stop missing

Line 38: a and it missing 

Line 43: in the begging year gap missing and last sentence reduce gap

Line 44: differences

LIne46: remove the

Line47: focus word is missing and in the end needs full stop.

Line 48: full form DM is missing

Line50-54: rewrite the sentence in your own word from an analysis of work from Svensson et al

Line56-59: This statement is mentioned in Siddiqui et al 2013 paper. However, the last statement in reference to women's prevalence is described in Raum et al 2012. Please mention this citation and describe the sentence clearly by giving the background of the medication adherence study.

Line 62: keep consistency in writing T1DM
Line68: HLA full form missing

Line76: no fullstop and random space 

Line78: space missing between paragraph and subheading

Line 81-83: rewrite in your own word from Siddiqui et al

Line 87:thy bask??

Line 88:: no full stop

Line 89: There is no connection between above statement from line 88 to this. Please mention a new subheading or make a new paragraph with a relevant statement.

Line91to 100: unnecessary sentences, write scientifically relevant to diabetes disease. grammar mistakes and typo

Line 101: grammatical wrong sentence

Line102- 108: write a statement as per your interpretation from magkos et al

Line 108-111: a statement made and contradictory statement in another paragraph that type2 cannot be translated to type 1

Line112-113: the sentence is not properly constructed grammar check

Line116-117: however, insulin also plays role in the inhibition of. please write sentences grammatically right

Line119: remove space and no paragraph formatting

Line 120-121: no mention of isotope 

Line125: grammar mistake in the sentence

Line131: contradictory statement 2 from ‘’However most of the data derive from studies in type 2 post-menopausal women and cannot be easily transposed to type 1 DM’’ clarify your remove

Line132: need space between sentences starting progesterone

Line134:gap

Line135: remove bracket and gap

Line138-139: no full stop and is it intentionally made in italic or accidentally so please change it and mention the full form of CSII

Line140-142: Reconstruct sentence properly. e.g   Finally overall data available shows that /confirm that

Line 143-144: which stimuli mention

Line145: VLDL full not mentioned

Line146: sentence does not make sense give a supporting description to this statement

Line147: This sentence needs to be rewritten properly with grammatical error

Line151: Is this statement in regards to insulin action /resistance  or something else

Line154: gap 

Line159: remove able and start from to create

Line162-164: The discussion is too short and does not justify with review titel

References

Please organize references properly with some software in the proper format e.g AMA, APA, MLA

Author Response

We changed the manuscript according to the suggestions of the reviewers. Starting from the title that now includes the definition of diabetes mellitus. Please see the attachment.

Reviewer 2 Report

Singh and Tatti higlight the lack of and importance of characterizing gender differences in medicine, focusing almost exclusively on Type 1 Diabetes. While this is an impactful area for review, at times the text is difficult to follow. Improvements should be made to the focus, writing and structure of the article.

The title suggests a broad overview on the topic of gender differences in medicine but as the review focuses on Type 1 Diabetes this is misleading and should be changed. The authors should make clear in the title that Type 1 Diabetes will be mainly discussed. 

The abstract itself is written in a confusing manner and highly emotive or opinionated sentences are used which make it difficult to decipher facts from the authors opinion, e.g. line 14 and line 17. The ending of the abstract is abrupt and could do with a concluding or leading sentence focusing on the conclusion, recommendation or benefit of the review to the field.

Frequently statements are made with no literature cited. This should be rectified, especially in the case of a review where the reader seeks to have a thorough introduction into the field and be exposed to important scientific findings that they can investigate further if desired. Examples where citations are missing are lines 32-34, 36, 62-63.

A brief introduction into type I diabetes would be beneficial to the reader and could possibly add continuity to later sections, for example the beginning of the "Incidence" section (lines 42-44) seems sudden and can be confusing to the reader (e.g. incidence of what?).

Common themes should be summarized where possible, e.g. in lines 42-54 the authors describe two studies where the incidence of Type 1 diabetes in males was higher than in females. However, they list these separately in a disjointed manner making it confusing for the reader to grasp. Introductory or summarizing sentences would be useful to preface and connect the results from multiple studies and allow the review to flow better.

In line 63, the authors mention that type 1 diabetes is often diagnosed during childhood. Could the time of diagnosis (before puberty) contribute to the lack of gender focused research? If so, this may be an interesting point to focus on.

The cut-off of 15 years of age is made in line 73 when looking at incidence rate. Could the authors explain why this cut-off was used and provide a citation?

Although the last paragraph is forward looking, it is very brief. It would be beneficial for the authors to expand on the focus and suggestions for future research. Similarly, are there any diseases where utilizing gender differences enhance diagnosis or therapy? What are the lessons learned there which could be applied to Type 1 Diabetes?

Author Response

We  changed the manuscript according to the suggestions of the reviewers. Please see the attachment.

Round 2

Author Response

Thank you for your comments.

Reviewer 2 Report

The manuscript is much improved and the authors have addressed many of my concerns. The review could still benefit from some minor edits. Particularly, emotive and leading language in the abstract remains, In lines 13-14 which infer a lack of motive on the drug companies part which is impossible to tell without reference. The authors could instead refer to this in the conclusion by pondering on the complications this research could cause to drug companies in respect to the applicability of their product for the general population (and use references to support this hypothesis). 

Line 50, the Incidence section still seems to start abruptly. Maybe a reference of Type I diebetes in the first sentence of the Incidence section could help here?

Author Response

Thank you for your comments.
